# SUBSTRUCTURED GRAPH CONVOLUTION FOR NON-OVERLAPPING GRAPH DECOMPOSITION

## ABSTRACT

Graph convolutional networks have been widely used to solve the graph problems such as node classification, link prediction, and recommender systems. It is well known that large graphs require large amount of memory and time to train graph convolutional networks. To deal with large graphs, many methods are being done, such as graph sampling or decomposition. In particular, graph decomposition has the advantage of parallel computation, but information loss occurs in the interface part. In this paper, we propose a novel substructured graph convolution that reinforces the interface part lost by graph decomposition. Numerical results indicate that the proposed method is robust in the number of subgraphs compared to other methods.

## 1 INTRODUCTION

Graph convolutional networks (GCNs) (Kipf & Welling, 2017) are widely used in node classification (Xiao et al., 2022), link prediction (Zhang & Chen, 2018), and recommender systems (Wu et al., 2022). For a given graph, GCN constructs a renormalized graph Laplacian using the graph's adjacency matrix and uses it for layer propagation. Therefore, as the dimension of the adjacency matrix of the graph increases, more memory and time are required to train the network.

There are two main types of research to solve the memory problem. The first is graph sampling methods (Hamilton et al., 2017; Chen et al., 2018; Ye et al., 2019; Zeng et al., 2020). These methods basically create a subgraph at every iteration using an appropriate sampling algorithm like Deep-Walk (Perozzi et al., 2014). The network is trained using this subgraph. GraphSAGE (Hamilton et al., 2017) used the edge information corresponding to a fixed-size neighborhood of uniformly sampled nodes. FastGCN (Chen et al., 2018) proposed the importance sampling and showed faster learning speed compared to GraphSAGE. VR-GCN (Ye et al., 2019) used the variance reduction technique to reduce the number of sampling nodes. GraphSAINT (Zeng et al., 2020) improved performance by using graph sampling instead of node sampling or edge sampling. Because the graph sampling method uses subgraphs to reduce memory usage, it is important to determine the number of samples. The higher the number of samples, the higher the performance is expected, but the slower the training speed and the memory is consumed.

On the one hand, there is another approach to decompose the graph (Chiang et al., 2019). The biggest advantage of the decomposition methods is that, unlike the sampling methods, it can be performed in advance before network training. A lot of research has been done on how to decompose the graph (Karypis & Kumar, 1998; Avery, 2011; Gonzalez et al., 2012). Among them, METIS (Karypis & Kumar, 1998), which can quickly decompose a graph using a multi-level structure, is widely used. In view of linear algebra, METIS derives a block diagonal matrix by performing a non-overlapping decomposition on the adjacency matrix of a given graph. ClusterGCN (Chiang et al., 2019) trains the network with a mini-batch gradient descent algorithm by performing block sampling on the block diagonal matrix generated by METIS. That is, this method trains the network by alternating block submatrices through random sampling. On the other hand, there is another way to train the network at once with the gradient descent algorithm by computing the block diagonal matrix for each block in parallel. A big difference from the alternating method is that it does not require inner iteration because it trains the network using all subgraphs at once and then merges them. However, non-overlapping decomposition drops blocks in off-diagonal part and does not supplement

information about this part. Therefore, as the number of blocks increases, the amount of information lost increases, which also affects training of the network.

In the field of numerical analysis, there are substructuring methods (Bramble et al., 1986; Farhat & Roux, 1991) that additionally use information on the interface part in the domain that has undergone non-overlapping decomposition. Assuming that the interface part is sparse when appropriate non-overlapping decomposition is performed, the added computation and communication costs are very small. Therefore, although the interface part requires sequential computation, it does not become a bottleneck in the overall parallel structure.

Motivated by the substructuring method, we modify the graph convolution with the block diagonal adjacency matrix generated by non-overlapping decomposition. That is, a substructure using the interface adjacency matrix is added to the graph convolution. We call a graph convolution with this added substructure a substructured graph convolution. A simple linear algebra calculation shows that the sum of the outputs of the aggregate using the block diagonal adjacency matrix and the interface adjacency matrix is different from the output of the aggregate using the original adjacency matrix. Therefore, to compensate for this difference, a weighted sum is performed by calculating coefficients by referring to the attention module that shows good performance in natural language processing (Vaswani et al., 2017) and image classification (Hu et al., 2018). From the numerical results, it can be confirmed that the proposed graph convolution adequately complements the interface part.

The rest of this paper is organized as follows. In Section 2, we introduce an abstract non-overlapping graph decomposition framework and two methods for training a given network with decomposed graphs. We present the substructured graph convolution in Section 3. Improved node classification accuracy or F1-score of the proposed graph convolution applied to GCN, GCNII, GAT, and SGC using various datasets is presented in Section 4. We conclude this paper with remarks in Section 5.

## 2 NON-OVERLAPPING GRAPH DECOMPOSITION

In this section, we briefly introduce an algebraic framework of non-overlapping graph decomposition. We then describe two methods for training the graph convolutional networks using the decomposed graphs.

### 2.1 ALGEBRAIC FRAMEWORK

Let $\boldsymbol{A} \in \mathbb{R}^{n \times n}$ be an adjacency matrix of a given graph consisting of $n$ nodes. Without loss of generality, let the graph be uniformly decomposed so that each subgraph has $n/N$ nodes for a positive integer $N$. Let $\boldsymbol{R}_k \colon \mathbb{R}^n \to \mathbb{R}^{n/N}$ be the restriction operator onto $k$-th subgraph.

We construct a non-overlapping decomposition of given adjacency matrix $\boldsymbol{A}$ under the node decomposition setting. A subgraph adjacency matrix $\boldsymbol{A}_k \in \mathbb{R}^{n/N \times n/N}$ is defined by

$$\boldsymbol{A}_k = \boldsymbol{R}_k \boldsymbol{A} \boldsymbol{R}_k^T, \quad k = 1, \cdots, N. \tag{2.1}$$

The non-overlapping decomposition $\widetilde{\boldsymbol{A}}$ of $\boldsymbol{A}$ with subgraph adjacency matrices (2.1) is given by

$$\widetilde{\boldsymbol{A}} = \sum_{k=1}^N \boldsymbol{R}_k^T \boldsymbol{A}_k \boldsymbol{R}_k. \tag{2.2}$$

Then $\widetilde{\boldsymbol{A}}$ becomes the adjacency matrix of the graph consisting of subgraphs having $\boldsymbol{A}_1, \cdots, \boldsymbol{A}_N$ as adjacency matrices. We define this graph as a non-overlapping graph decomposition for a given graph.

For the block matrix representation

$$\boldsymbol{A} = [\boldsymbol{A}_{ij}]_{1 \leq i,j \leq N} = \begin{bmatrix} \boldsymbol{A}_{11} & \boldsymbol{A}_{12} & \ldots & \boldsymbol{A}_{1N} \\ \boldsymbol{A}_{22} & \boldsymbol{A}_{22} & \ldots & \boldsymbol{A}_{2N} \\ \vdots & \vdots & \ddots & \vdots \\ \boldsymbol{A}_{N1} & \boldsymbol{A}_{N2} & \ldots & \boldsymbol{A}_{NN} \end{bmatrix}, \tag{2.3}$$

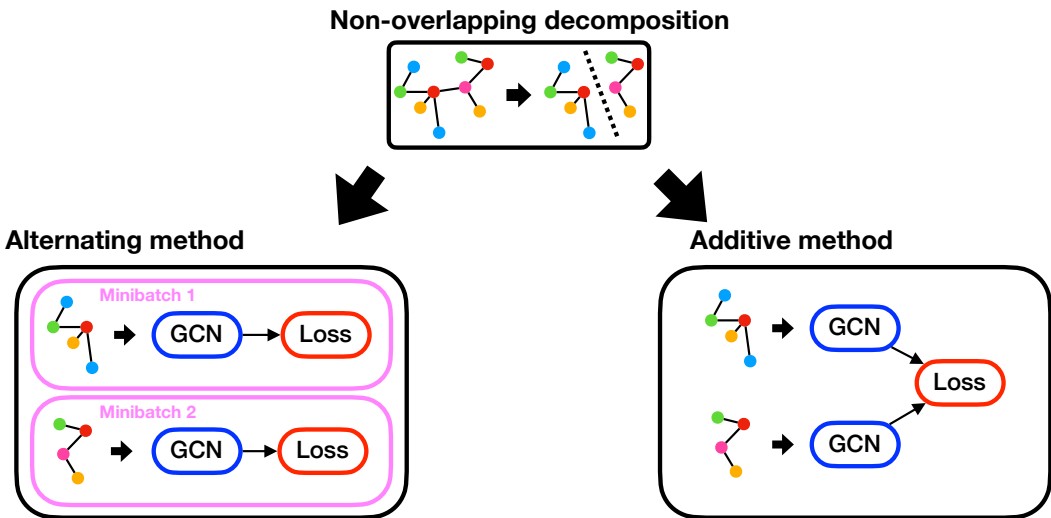

Figure 1: Schematic description of the alternating method and the additive method after non-overlapping graph decomposition. Note that we assume the case of $N = 2$ for simplicity.

the corresponding block matrix representation of the non-overlapping decomposition $\widetilde{A}$ is written as

$$\widetilde{A} = \mathrm{diag}\left([A_{ii}]_{i=1}^N\right) = \begin{bmatrix} A_{11} & 0 & \dots & 0 \\ 0 & A_{22} & \dots & 0 \\ \vdots & \vdots & \ddots & \vdots \\ 0 & 0 & \dots & A_{NN} \end{bmatrix}, \tag{2.4}$$

where $A_{ij} \in \mathbb{R}^{n/N \times n/N}$ is defined by $A_{ij} = R_i A R_j^T$. That is, $\widetilde{A}$ is the block-diagonal part of $A$. Comparing (2.3) and (2.4), it can be seen that the number of off-diagonal parts lost in $A$ increases as $N$ increases. That is, it is clear that as $N$ increases, the degree to which $\widetilde{A}$ approximates $A$ decreases significantly; see Toselli & Widlund (2005).

## 2.2 TRAINING WITH NON-OVERLAPPING GRAPH DECOMPOSITION

Now, we introduce two methods for training a network with a graph generated by the non-overlapping graph decomposition. One is an alternating method known as ClusterGCN (Chiang et al., 2019) and the other is an additive method similar to Data Parallelism (Gonzalez et al., 2012).

First, we explain the basic training method of graph convolutional networks. Let $\mathcal{G} = (V, A)$ be a graph consisting of node vector $V = (v_1, \cdots, v_n)$ with an adjacency matrix $A \in \mathbb{R}^{n \times n}$. Each node $v_i$ has an $F$-dimensional feature vector $x_i \in \mathbb{R}^F$ and belongs to one of the $C$ classes, which is labeled with a $C$-dimensional one-hot vector $y_i$. The entire node feature $X \in \mathbb{R}^{n \times F}$ has $\{x_1, \cdots, x_n\}$ as row vectors. Similarly, the entire node label $Y \in \mathbb{R}^{n \times C}$ has $\{y_1, \cdots, y_n\}$ as row vectors.

For convenience, it is assumed that the network $f_\Theta$ consists of one graph convolution layer with trainable parameter $\Theta$ and the softmax function. Let $W \in \mathbb{R}^{F \times C}$ and $b \in \mathbb{R}^C$ be the weight and bias of the layer and $\Theta = \{W, b\}$. Then the forward propagation of $f_\Theta$ is written as

$$f_\Theta(x, A) = \mathrm{softmax}(SXW + b),$$

where $S = (I + D)^{-\frac{1}{2}}(I + A)(I + D)^{-\frac{1}{2}}$ is a renormalized graph Laplacian (Kipf & Welling, 2017). Here $D$ is a degree matrix of given $A$ and $I$ is an identity matrix. Let $\mathcal{L}$ be a loss function that trains the network $f_\Theta$. Then, for each epoch, the network is trained by gradient descent method

$$\Theta^{j+1} = \Theta^j - \eta \nabla_{\Theta^j} \mathcal{L}(f_{\Theta^j}(X, A), Y).$$

Now, we introduce the alternating method first. Using (2.1) and (2.2), each subgraph $\mathcal{G}_k$, feature matrix $\boldsymbol{X}_k$ and label matrix $\boldsymbol{Y}_k$ are defined as

$$\mathcal{G}_k = \{\boldsymbol{V}_k, \boldsymbol{A}_k\}, \ \boldsymbol{V}_k = \boldsymbol{R}_k \boldsymbol{V}, \ \boldsymbol{X}_k = \boldsymbol{R}_k \boldsymbol{X}, \ \boldsymbol{Y}_k = \boldsymbol{R}_k \boldsymbol{Y}.$$

Then, for each epoch, the network is trained by mini-batch gradient descent method described in Algorithm 1.

---

**Algorithm 1:** The alternating method with learning rate $\eta$

---

**for** $k = 1, \cdots, N$ **do**
$\quad \mid \quad \Theta^{j+1} = \Theta^j - \eta \nabla_{\Theta^j} \mathcal{L}(f_{\Theta^j}(\boldsymbol{X}_k, \boldsymbol{A}_k), \boldsymbol{Y}_k)$
**end**

---

Next, we introduce the additive method. The additive method computes the total output of layer gathering the outputs derived from each subgraph in parallel. After that, the network is trained using the gradient descent method. For each epoch, Algorithm 2 shows the update process of additive method.

---

**Algorithm 2:** The additive method with learning rate $\eta$

---

**for** $k = 1, \cdots, N$ **in parallel do**
$\quad \mid \quad \widetilde{\boldsymbol{Y}}_k = f_{\Theta^j}(\boldsymbol{X}_k, \boldsymbol{A}_k)$
**end**
$\widetilde{\boldsymbol{Y}} = \sum_{k=1}^{N} \boldsymbol{R}_k^T \widetilde{\boldsymbol{Y}}_k$
$\Theta^{j+1} = \Theta^j - \eta \nabla_{\Theta^j} \mathcal{L}(\widetilde{\boldsymbol{Y}}, \boldsymbol{Y})$

---

Figure 1 illustrates a schematic description of the additive method and the alternating method in the case of $N = 2$.

## 3 SUBSTRUCTURED GRAPH CONVOLUTION

As mentioned in Section 2, the non-overlapping graph decomposition has a disadvantage in that the loss of off-diagonal information of the adjacency matrix increases as $N$ increases. Therefore, it can be expected that the performance of graph convolutional networks using graph decomposition depends heavily on the number of subgraphs $N$. The same phenomenon can be observed in both the alternating method and the additive method, and the experiment for this will be discussed in Section 4.

### 3.1 ALGEBRAIC FRAMEWORK

In numerical analysis, there is a *substructuring* method that increases performance by using the interface part without disturbing the parallel structure; see, e.g., Toselli & Widlund (2005); Dolean et al. (2015). Let the non-overlapping decomposition $\widetilde{\boldsymbol{A}}$ defined in Section 2 be the interior adjacency matrix. We consider an interface adjacency matrix $\widehat{\boldsymbol{A}} = \boldsymbol{A} - \widetilde{\boldsymbol{A}}$. Then, we propose a novel graph convolution called *substructured graph convolution*, which improves performance by adding interface parts like the substructuring method, while maintaining parallel structure. We now define the renormalized graph Laplacian $\widetilde{\boldsymbol{S}}$ and $\widehat{\boldsymbol{S}}$ for $\widetilde{\boldsymbol{A}}$ and $\widehat{\boldsymbol{A}}$, respectively, as

$$\widetilde{\boldsymbol{S}} = (\boldsymbol{I} + \widetilde{\boldsymbol{D}})^{-\frac{1}{2}}(\boldsymbol{I} + \widetilde{\boldsymbol{A}})(\boldsymbol{I} + \widetilde{\boldsymbol{D}})^{-\frac{1}{2}},$$
$$\widehat{\boldsymbol{S}} = (\boldsymbol{I} + \widehat{\boldsymbol{D}})^{-\frac{1}{2}}(\boldsymbol{I} + \widehat{\boldsymbol{A}})(\boldsymbol{I} + \widehat{\boldsymbol{D}})^{-\frac{1}{2}},$$

where $\widetilde{\boldsymbol{D}}$ and $\widehat{\boldsymbol{D}}$ are the degree matrices of $\widetilde{\boldsymbol{A}}$ and $\widehat{\boldsymbol{A}}$, respectively. Note that $\boldsymbol{S} \neq \widetilde{\boldsymbol{S}} + \widehat{\boldsymbol{S}}$. In other words, simply adding the outputs of graph convolution using $\widetilde{\boldsymbol{S}}$ and $\widehat{\boldsymbol{S}}$ is different from the output of graph convolution using $\boldsymbol{S}$, the renormalized Laplacian matrix for $\boldsymbol{A}$. Therefore, we consider a weighted sum of $\widetilde{\boldsymbol{S}}$ and $\widehat{\boldsymbol{S}}$ rather than simple addition to get a better approximation to $\boldsymbol{S}$.

Figure 2: Graphical description of the proposed substructured graph convolution after non-overlapping graph decomposition. Note that we assume the case of $N = 2$ for simplicity.

Table 1: The number of edge-cut according to the number of subgraphs $N$ with random, ordered, and METIS decomposition applied to Cora. The random decomposition method uses the randperm function in PyTorch, and the ordered decomposition method divides the node indices in order of Cora. Note that the total number of edges in the Cora is $5429$.

| N | Random | Ordered | METIS |
|---|--------|---------|-------|
| 2 | 2613 | 2603 | 186 |
| 4 | 3907 | 3682 | 351 |
| 8 | 4580 | 4340 | 537 |
| 16 | 4898 | 4649 | 714 |
| 32 | 5082 | 4817 | 1018 |
| 64 | 5187 | 4941 | 1313 |

With appropriate coefficient vectors $\widetilde{\boldsymbol{\alpha}}$ and $\widehat{\boldsymbol{\alpha}}$, we consider the forward propagation of substructured graph convolution $f_\Theta$ as

$$f_\Theta(\boldsymbol{X}, \widetilde{\boldsymbol{A}}, \widehat{\boldsymbol{A}}) = \sigma(\{\mathrm{diag}(\widetilde{\boldsymbol{\alpha}})\widetilde{\boldsymbol{S}} + \mathrm{diag}(\widehat{\boldsymbol{\alpha}})\widehat{\boldsymbol{S}}\}\boldsymbol{X}\boldsymbol{W} + \boldsymbol{b}),$$

where $\Theta = \{\boldsymbol{W}, \boldsymbol{b}\}$ is a parameter of the layer and $\sigma$ is a nonlinear activation function.

For parallel computation of the substructured graph convolution, information compression is required to minimize communication between the interior part $\widetilde{\boldsymbol{S}}$ and the interface part $\widehat{\boldsymbol{S}}$ in designing $\widetilde{\boldsymbol{\alpha}}$ and $\widehat{\boldsymbol{\alpha}}$. Motivated by the attention module, which is the core of the SE block used in CNN (Hu et al., 2018), and the transformer structure mainly used in NLP (Vaswani et al., 2017), we consider the coefficients $\widetilde{\boldsymbol{\alpha}}$ and $\widehat{\boldsymbol{\alpha}}$ such as

$$[\widetilde{\boldsymbol{\alpha}}, \widehat{\boldsymbol{\alpha}}] = \mathrm{softmax}\left(\left[\frac{1}{F}\sum_{i=1}^{F}(\widetilde{\boldsymbol{S}}\boldsymbol{X})_i, \frac{1}{F}\sum_{i=1}^{F}(\widehat{\boldsymbol{S}}\boldsymbol{X})_i\right]\right),$$

where $(\cdot)_i$ denotes the $i$-th column. Note that $F$ is the feature dimension of $\boldsymbol{X}$. This operation compresses the information of the intermediate features generated by $\widetilde{\boldsymbol{S}}$ and $\widehat{\boldsymbol{S}}$, and then obtains the softmax value with minimal communication and computation cost. The process of computing the coefficients $\widetilde{\boldsymbol{\alpha}}$ and $\widehat{\boldsymbol{\alpha}}$ is a structure that needs sequential computation, but it does not need additional parameters and uses the minimum cost to solve the bottleneck in the overall parallel structure. A brief graphical description of the substructured graph convolution for the case of $N = 2$ is shown in Figure 2.

## 3.2 IMPLEMENTATION ISSUES

In this section, we discuss several issues on efficient implementation of the proposed substructured graph convolution. The first is the selection of an algorithm that performs the non-overlapping decomposition to a given graph. It is natural that the density of the interface adjacency matrix increases

Table 2: Details of used datasets. Cora, CiteSeer, and PubMed datasets have a single class label, but the PPI dataset can have multiple class labels.

| Dataset | Nodes | Edges | Features | Classes | Train / Validation / Test |
|---------|-------|-------|----------|---------|---------------------------|
| Cora | 2708 | 5429 | 1433 | 7 | 140 / 500 / 1000 |
| CiteSeer | 3327 | 4732 | 3703 | 6 | 120 / 500 / 1000 |
| PubMed | 19717 | 44338 | 500 | 3 | 60 / 500 / 1000 |
| PPI | 56944 | 818716 | 50 | 121 | 44906 / 6514 / 5524 |

according to the shape of the graph if it is simply randomly cut or divided in order. This makes, the interior adjacency matrix becomes sparser, which degrades the performance of the existing graph convolution. Note that as the interface adjacency matrix is sparse, the amount of sequential computation decreases, so that the proposed graph convolution can be computed more efficiently. For this reason, we need a non-overlapping graph decomposition algorithm that minimizes edge-cuts. METIS (Karypis & Kumar, 1998) is one of the good algorithms, that uses a multi-level structure to quickly perform the decomposition and minimize edge-cuts. Table 1 shows the number of edge-cuts according to each decomposition method applied to Cora (McCallum et al., 2000). METIS shows far fewer edge-cuts than simple random and ordered decomposition methods. Therefore, we use METIS for non-overlapping graph decomposition in the sequel.

Next, we discuss why the renormalized graph Laplacian $\widetilde{S}$ and $\widehat{S}$ are used. Since $A = \widetilde{A} + \widehat{A}$, the renormalized graph Laplacian $S$ can be decomposed as

$$S = (I + D)^{-\frac{1}{2}}(I + \widetilde{A})(I + D)^{-\frac{1}{2}} + (I + D)^{-\frac{1}{2}}\widehat{A}(I + D)^{-\frac{1}{2}}. \tag{3.1}$$

Therefore, when the decomposition (3.1) is performed, the computation of $(I + D)^{-\frac{1}{2}}$ is required for the computation of the interface part. This reduces the computational efficiency of the interface part that requires sequential computation and may cause a bottleneck in the interior part where parallel computation is possible. On the other hand, using $\widehat{S}$, sequential computation for the interface part can be efficiently performed.

Lastly, we note that the proposed substructured graph convolution was made by referring to the renormalized graph Laplacian of GCN, but it is also applicable to GCNII (Chen et al., 2020), GAT (Veličković et al., 2018), and SGC (Wu et al., 2019). The key idea is to construct the interface adjacency matrix, process the aggregate, and then determine each coefficient via the attention module. Then, substructured graph convolution is performed using the renormalized graph Laplacian corresponding to the aggregate part of a given network instead of the whole renormalized graph Laplacian.

## 4 APPLICATIONS

In this section, we present numerical results of the proposed graph convolution embedded into several existing GCNs: GCN, GCNII, GAT, and SGC. We evaluate the performance of transductive learning and inductive learning, which are mainly used as benchmarks in graph node classification problems.

For the transductive learning task, standard citation network benchmark datasets Cora (McCallum et al., 2000), CiteSeer (Giles et al., 1998), and PubMed (Yang et al., 2016) were used. In these datasets, a node and an edge mean a document and a citation, respectively. For the transductive environment, only 20 training nodes were used per class, and 500 and 1,000 nodes were used for validation and test, respectively. For the inductive learning task, we used a protein-protein interaction (PPI) dataset (Hamilton et al., 2017) consisting of graphs of different human tissues. The dataset has 20 training graphs and 2 validation and test graphs each. Also, the graph of the PPI dataset can have multiple class labels. Details of the number of nodes, edges, features, and classes in the dataset are given in Table 2.

Table 3: The accuracy of GCN, GCNII, GAT, and SGC on Cora, CiteSeer, and PubMed datasets.

| Network | Cora | CiteSeer | PubMed |
|---------|------|----------|--------|
| GCN | 82.40 | 71.60 | 78.90 |
| GCNII | 83.40 | 73.40 | 78.40 |
| GAT | 81.90 | 71.30 | 78.80 |
| SGC | 79.80 | 72.00 | 76.90 |

All networks were implemented in Python with PyTorch (Paszke et al., 2019) and PyG (Fey & Lenssen, 2019), and all computations were performed on a cluster equipped with Intel Xeon Gold 6240R (2.4GHz, 48C) CPUs, NVIDIA 3090 GPUs, and operating system CentOS 7.8.

## 4.1 TRANSDUCTIVE LEARNING

Transductive learning is a type of semi-supervised learning, i.e., given the nodes and edges of the graph, the network is trained using the labeled nodes, and then the unlabeled nodes are labeled. In particular, the transductive learning uses the same graph for training and testing. Thus, the transductive task shows how much the adjacency matrix of a given graph affects the labeling performance of the network.

### 4.1.1 NETWORK AND HYPERPARAMETER SETUP

GCN is a two-layer model which has 16 channels for the Cora and CiteSeer datasets and 64 channels for the PubMed dataset. GCNII is a model using 64 layers with 64 channels for the Cora and CiteSeer datasets and 16 layers with 256 channels for the PubMed dataset. Note that the hyperparameters $\alpha$ and $\lambda$ for GCNII are 0.1 and 0.5, respectively. GAT is a model using two layers with 8 headers each with 8 channels. The last layer of GAT averages the outputs of the headers and all other layers concatenate the outputs. Finally, SGC performs feature propagation twice. Note that GAT uses ELU (Clevert et al., 2015) and other networks use ReLU as the activation function.

All neural networks were trained for 200 epochs using Adam optimizer (Kingma & Ba, 2014). The learning rate, weight decay, and dropout (Srivastava et al., 2014) were determined to give the best performance through grid search at $[0.001, 0.005, 0.01]$, $[0, 0.0001, 0.0005]$, and $[0, 0.4, 0.6, 0.8]$, respectively.

### 4.1.2 EXPERIMENT RESULTS

First, to verify the performance of each convolution, we provide Table 3, which shows the accuracy of given networks trained using the standard graph convolution for standard citation network benchmark datasets. Next, we compare the performance of the proposed substructured graph convolution with the additive method and the alternating method. Table 4 shows numerical results of all of the previously mentioned methods applied to GCN, GCNII, GAT, and SGC with Cora, CiteSeer, and PubMed datasets. First of all, as $N$ increases, the overall accuracy decreases regardless of the network and dataset. In particular, in the cases of $N = 32$ and 64, it can be seen that the accuracy of the additive and alternating methods is much lowered because the edge-cuts of the given graph are very large. On the other hand, the substructured graph convolution shows that the decrease in accuracy is small as $N$ increases compared to the additive and alternating methods. Moreover, in certain cases, substructured graph convolution shows better performance than standard graph convolution. This shows that adding the information of the interface part is properly reflected in the network and helps to improve accuracy.

## 4.2 INDUCTIVE LEARNING

The inductive learning task is a supervised learning. The biggest difference from the transductive learning is that the graphs for testing are different from the training graphs. If the adjacency matrix of the training graph is block diagonal, the trained network learns the block diagonal shape, so it is

Table 4: The accuracy of GCN, GCNII, GAT, and SGC equipped with additive (AD), alternating (AL), and substructuring (SS) for the Cora, CiteSeer, and PubMed datasets, where $N$ denotes the number of subgraphs.

| Network | Type | N | Cora | CiteSeer | PubMed | Network | Type | N | Cora | CiteSeer | PubMed | Network | Type | N | Cora | CiteSeer | PubMed |
|---|---|---|---|---|---|---|---|---|---|---|---|---|---|---|---|---|---|
| GCN | AD | 2 | **80.60** | 71.00 | 79.00 | GCN | AD | 4 | **79.90** | 71.40 | 78.90 | GCN | AD | 8 | 79.20 | 71.60 | 77.70 |
|  | AL | 2 | 80.00 | 71.40 | **79.10** |  | AL | 4 | 78.10 | 71.50 | **79.10** |  | AL | 8 | **80.30** | 71.90 | **78.20** |
|  | SS | 2 | 80.30 | **72.50** | 78.30 |  | SS | 4 | 78.90 | **72.20** | 78.30 |  | SS | 8 | 79.90 | **72.00** | 78.00 |
| GCNII | AD | 2 | 82.30 | 70.30 | 78.40 | GCNII | AD | 4 | 81.00 | 72.80 | 77.00 | GCNII | AD | 8 | 82.60 | 73.20 | 76.30 |
|  | AL | 2 | 82.10 | 70.20 | 77.10 |  | AL | 4 | 80.90 | 70.60 | 78.20 |  | AL | 8 | **83.20** | 69.00 | 78.00 |
|  | SS | 2 | **83.60** | **72.20** | **79.30** |  | SS | 4 | **82.60** | **74.00** | **79.30** |  | SS | 8 | 82.20 | **73.90** | **81.00** |
| GAT | AD | 2 | 81.30 | **70.90** | **78.40** | GAT | AD | 4 | 79.50 | 69.90 | 78.50 | GAT | AD | 8 | 80.80 | 69.00 | 76.70 |
|  | AL | 2 | 80.70 | 70.10 | 78.10 |  | AL | 4 | 80.20 | 70.60 | 78.10 |  | AL | 8 | **82.00** | 68.30 | 78.10 |
|  | SS | 2 | **81.50** | 69.30 | 78.30 |  | SS | 4 | **81.60** | **70.70** | **78.70** |  | SS | 8 | 81.90 | **70.60** | **78.30** |
| SGC | AD | 2 | 78.80 | 71.70 | 76.30 | SGC | AD | 4 | 78.90 | 70.40 | 77.20 | SGC | AD | 8 | **79.30** | 71.30 | 76.50 |
|  | AL | 2 | 74.30 | 71.30 | 75.20 |  | AL | 4 | 73.60 | 61.70 | 75.50 |  | AL | 8 | 66.80 | 23.70 | 75.40 |
|  | SS | 2 | **79.30** | **72.00** | **77.20** |  | SS | 4 | **79.70** | **71.90** | **77.60** |  | SS | 8 | 79.10 | **72.20** | **78.70** |

| Network | Type | N | Cora | CiteSeer | PubMed | Network | Type | N | Cora | CiteSeer | PubMed | Network | Type | N | Cora | CiteSeer | PubMed |
|---|---|---|---|---|---|---|---|---|---|---|---|---|---|---|---|---|---|
| GCN | AD | 16 | 80.10 | 71.10 | 77.40 | GCN | AD | 32 | 79.10 | 70.10 | 78.40 | GCN | AD | 64 | 78.40 | 67.50 | 78.10 |
|  | AL | 16 | **81.40** | 72.10 | 77.30 |  | AL | 32 | 78.10 | 71.50 | **78.60** |  | AL | 64 | 76.10 | 70.50 | 76.80 |
|  | SS | 16 | 80.20 | **72.20** | **79.00** |  | SS | 32 | **80.20** | **72.80** | **78.60** |  | SS | 64 | **81.20** | **72.00** | **78.60** |
| GCNII | AD | 16 | 80.40 | **72.70** | 75.20 | GCNII | AD | 32 | 81.60 | **73.00** | 74.80 | GCNII | AD | 64 | 79.30 | 71.00 | 76.00 |
|  | AL | 16 | 81.60 | 70.00 | 77.20 |  | AL | 32 | 81.50 | 58.40 | 76.70 |  | AL | 64 | 78.30 | 45.30 | 77.60 |
|  | SS | 16 | **84.60** | **72.70** | **80.90** |  | SS | 32 | **83.50** | **73.00** | **80.20** |  | SS | 64 | **83.80** | **75.30** | **81.10** |
| GAT | AD | 16 | 80.40 | 70.70 | 77.50 | GAT | AD | 32 | 77.70 | 69.90 | 77.80 | GAT | AD | 64 | 77.10 | 69.20 | 77.20 |
|  | AL | 16 | 78.90 | 70.00 | 76.80 |  | AL | 32 | 79.70 | 68.80 | 78.90 |  | AL | 64 | 78.30 | 69.90 | 77.30 |
|  | SS | 16 | **81.10** | **70.90** | **80.00** |  | SS | 32 | **82.20** | **70.00** | **79.10** |  | SS | 64 | **83.00** | **70.60** | **79.10** |
| SGC | AD | 16 | 77.80 | 71.90 | 75.90 | SGC | AD | 32 | 77.60 | 71.50 | 76.20 | SGC | AD | 64 | 77.40 | 70.50 | 76.00 |
|  | AL | 16 | 74.10 | 37.90 | 67.60 |  | AL | 32 | 72.90 | 47.60 | 75.60 |  | AL | 64 | 61.40 | 65.00 | 71.90 |
|  | SS | 16 | **79.50** | **72.80** | **78.70** |  | SS | 32 | **79.70** | **72.50** | **78.40** |  | SS | 64 | **80.40** | **72.10** | **78.80** |

difficult to expect labeling performance for a general graph. Thus, the inductive learning task shows the effect of reflecting the interface information on the generalizability of the network.

### 4.2.1 NETWORK AND HYPERPARAMETER SETUP

GCN is a three-layer model which has 1,024-channel layers with skip-connections (He et al., 2016). GCNII is a model using 9 layers with 2,048 channels. Note that the hyperparameters $\alpha$ and $\lambda$ for GCNII are set to 0.5 and 1.0, respectively. GAT is a model using three layers with 4 headers each with 256 channels. Also, the skip-connection is used in GAT. Finally, SGC performs feature propagation three times. GCN and GAT used ELU as the activation function, and the rest used ReLU. The optimizer and hyper parameters for training were set in the same way as in Section 4.1. In addition, the network was trained using a total of 20 PPI training graphs, one at a time, and the sequence of training graphs is shuffled every epoch.

### 4.2.2 EXPERIMENT RESULTS

Similar to the transductive learning task, we compare the proposed substructured graph convolution with the additive and alternating methods. Note that F1-scores for the PPI datasets of GCN, GCNII, GAT, and SGC trained using the standard graph convolution are 99.06, 89.79, 99.33, and 76.10, respectively. The numerical results in Table 5 confirm that the proposed substructured graph convolution performs better than other methods even on inductive learning tasks. In particular, the other two methods show that the F1-score drops sharply as $N$ increases, whereas the substructured graph convolution shows little change in the F1-score. In other words, it can be seen that the form of the adjacency matrix of the graph used for training is important for the generalizability of the network. The adjacency matrix used in the additive method and the alternating method is in the form of a block diagonal excluding the interface part, and it can be seen that the interface adjacency matrix plays a large role in the generalizability of the network. In addition, it can be confirmed that the proposed substructured graph convolution improves the performance of the network by using the interface adjacency matrix appropriately in the inductive learning task as in the transductive learning task.

## 5 CONCLUSION

In this paper, we proposed a novel substructured graph convolution that is suitable for parallel computation and robust with respect to large numbers of subgraphs. Since the additional structure re-

Table 5: F1-scores of GCN, GCNII, GAT, and SGC equipped with additive (AD), alternating (AL), and substructuring (SS) for the PPI dataset, where $N$ is the number of subgraphs.

| Network | Type | N | PPI | Network | Type | N | PPI | Network | Type | N | PPI |
|---|---|---|---|---|---|---|---|---|---|---|---|
| GCN | AD | 2 | 97.97 | GCN | AD | 4 | 94.34 | GCN | AD | 8 | 91.70 |
|  | AL | 2 | **98.17** |  | AL | 4 | 95.10 |  | AL | 8 | 92.49 |
|  | SS | 2 | 95.68 |  | SS | 4 | **96.52** |  | SS | 8 | **96.87** |
| GCNII | AD | 2 | 87.78 | GCNII | AD | 4 | 85.00 | GCNII | AD | 8 | 91.70 |
|  | AL | 2 | **88.20** |  | AL | 4 | 84.89 |  | AL | 8 | **92.49** |
|  | SS | 2 | 87.29 |  | SS | 4 | **88.55** |  | SS | 8 | 88.76 |
| GAT | AD | 2 | 98.20 | GAT | AD | 4 | 93.48 | GAT | AD | 8 | 90.39 |
|  | AL | 2 | **98.42** |  | AL | 4 | 94.31 |  | AL | 8 | 91.13 |
|  | SS | 2 | 98.29 |  | SS | 4 | **98.66** |  | SS | 8 | **98.52** |
| SGC | AD | 2 | 75.29 | SGC | AD | 4 | 75.00 | SGC | AD | 8 | 74.83 |
|  | AL | 2 | **75.79** |  | AL | 4 | 75.07 |  | AL | 8 | 74.90 |
|  | SS | 2 | 75.05 |  | SS | 4 | **75.25** |  | SS | 8 | **75.26** |
| Network | Type | N | PPI | Network | Type | N | PPI | Network | Type | N | PPI |
| GCN | AD | 16 | 88.39 | GCN | AD | 32 | 85.96 | GCN | AD | 64 | 84.24 |
|  | AL | 16 | 89.68 |  | AL | 32 | 78.82 |  | AL | 64 | 73.88 |
|  | SS | 16 | **96.61** |  | SS | 32 | **96.18** |  | SS | 64 | **95.58** |
| GCNII | AD | 16 | 82.18 | GCNII | AD | 32 | 81.56 | GCNII | AD | 64 | 80.84 |
|  | AL | 16 | 79.78 |  | AL | 32 | 78.13 |  | AL | 64 | 76.43 |
|  | SS | 16 | **88.30** |  | SS | 32 | **88.16** |  | SS | 64 | **85.33** |
| GAT | AD | 16 | 86.72 | GAT | AD | 32 | 83.55 | GAT | AD | 64 | 81.62 |
|  | AL | 16 | 75.88 |  | AL | 32 | 72.73 |  | AL | 64 | 73.24 |
|  | SS | 16 | **98.38** |  | SS | 32 | **98.41** |  | SS | 64 | **98.49** |
| SGC | AD | 16 | 74.72 | SGC | AD | 32 | 74.65 | SGC | AD | 64 | 74.60 |
|  | AL | 16 | 74.83 |  | AL | 32 | 74.71 |  | AL | 64 | 74.68 |
|  | SS | 16 | **75.27** |  | SS | 32 | **75.20** |  | SS | 64 | **75.10** |

quires little computation and has no parameters, the proposed graph convolution does not cause a large bottleneck in parallel computation. We have experimentally shown that this novel graph convolution can train networks more effectively than additive and alternating methods. It also outperformed the standard graph convolution in certain cases. We expect that the proposed graph convolution can be efficiently utilized to train large graph datasets through multiple GPUs.

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
