# OpenReview forum: "Substructured Graph Convolution for Non-overlapping Graph Decomposition"
_ICLR.cc/2023/Conference — Submitted to ICLR 2023_

### Official Review · Reviewer_Tixk · 2022-10-23

**Confidence:** 4
**Correctness:** 3
**Technical Novelty And Significance:** 2
**Empirical Novelty And Significance:** 2
**Recommendation:** 3

**Clarity, Quality, Novelty And Reproducibility:**

As mentioned in the above section, though this paper presents clearly and has nice reproducibility, its novelty, technical depth, and experiment evaluation are a bit weak.

**Strength And Weaknesses:**

Strengths:

1. The implementation details are described clearly, providing better reproducibility.
2. The presentation is clear.


I have the following concerns about the paper:

1. The novelty of the proposed method, i.e., use METIS to partition the graph then training GNNs over them, is a bit limited. e.g., [1] ClusterGCN shares similar core ideas.
2. The technical depth is a bit limited. It would be better if the authors can provide better decomposition/partition method more suitable for GNN training, or provide better insights about the information loss or the usage of the `interfaces`.
3. Although in the abstract the authors stated the motivation is to deal with large graphs, the experiments are limited to small-size graphs.



[1] Cluster-GCN: An Efficient Algorithm for Training Deep and Large Graph Convolutional Networks, Wei-Lin Chiang, Xuanqing Liu, Si Si, Yang Li, Samy Bengio, Cho-Jui Hsieh, SIGKDD 2019

**Summary Of The Paper:**

This paper introduces how to train GNNs using non-overlapping decomposed graphs,  in order to scale GNNs to larger graphs and to obtain better performance.

**Summary Of The Review:**

In summary, the concerns about novelty, technical depths, and experiment evaluation outweigh the merits.

---

### Official Review · Reviewer_zYgm · 2022-10-24

**Confidence:** 4
**Correctness:** 2
**Technical Novelty And Significance:** 2
**Empirical Novelty And Significance:** 2
**Recommendation:** 3

**Clarity, Quality, Novelty And Reproducibility:**

The main benefit of training GCN on non-overlapping decomposed graph is to trading off model accuracy for less memory footprint and training time. The trade-off is unclear with SS method and is not well analysed or experimented. The memory footprint and training time benefits are supposed to be reduced significantly by the proposed SS method.

**Strength And Weaknesses:**

Strength:
  - The paper is well-written with good notations, figures, and pseudo codes.

Weaknesses:
  - The paper only compares result of the new method (SS) against other learning scheme on decomposed graph (AD and AL). Both AD and AL ignores the links between subgraphs, so it's obvious for SS to have prediction performance advantage. The paper does not study the increase in computation and memory cost though. Saving computation and memory cost is the main reason why graph decomposition is needed in the first place. I believe SS method reduces that benefit significantly.
  - It is unclear why the weighting formula is proposed that way. The author says to "minimal communication and computation cost" between subgraph and interface features, which I don't know what that actually means.

**Summary Of The Paper:**

The paper proposes a way to improve performance of GCN when decomposing graph. To reduce GCN complexity on large graphs, a graph can be decomposed into non-overlapping subgraphs. GCN is then can be trained only on these subgraphs, ignoring the "interface links" that connect between subgraphs. To reduce information loss due to ignoring the interface links, the paper proposes to train GCN on both subgraph and interface part (SS method). The output is then a weighted average of the node features obtained by subgraph and interface part.


**Summary Of The Review:**

The proposed SS method for training GCN on non-overlapping decomposed graph has unclear benefit.

---

### Official Review · Reviewer_TgT5 · 2022-10-26

**Confidence:** 2
**Correctness:** 2
**Technical Novelty And Significance:** 2
**Empirical Novelty And Significance:** 2
**Recommendation:** 3

**Clarity, Quality, Novelty And Reproducibility:**

The paper is generally easy to follow, although the problem could be better motivated, and some of the "system" assumptions could be explicitly stated. For example, do you assume that the system memory will be at least in the order of number of nodes, n?

The idea of utilizing "residual" graph component (after the decomposition), the so-called "interface adjacency matrix", for training is fairly straightforward, provided that the system memory is sufficiently large to hold the  "interface adjacency matrix". Otherwise, it seems to defeat the purpose.  I would have liked to see some analysis on the memory requirements and comparisons between your proposed method and other existing methods. Just because your method outperforms other methods on some datasets does not mean your method will be "practical". The trade-offs between performance vs. memory requirements, etc. should be analyzed and evaluated.

On an "intellectual" level, the proposed method is rather ad hoc. There is no attempt in making some mathematical justifications on the proposed "substructuring" method and GNN architectures used, for example, the specific forms of $\hat{S}$, and the way the learning based on this "graph structure" and how it is combined with the learning based on the decomposed graph components.

More generally, why is METIS is the right choice for graph decompostion, which is oblivious of node features on the graph? Why non-overlapping graph decomposition is the right choice?



**Details Of Ethics Concerns:**

No ethical concerns.

**Strength And Weaknesses:**

Strength:

  + Addressing scaling of GNN training on large graphs.


Weaknesses:
   - With the use of the so-called "interface adjacency matrix", albeit in general sparser than the original matrix, still requires O(n) memory to store it. If n is larger than the available  memory, the proposed method will not work, whereas the decomposition methods such as (Chiang et al., 2019) requires only O([n/N]^2) memory to store individual graph components.
   - The proposed method is rather ad hoc. There is no attempt in mathematically justification the proposed method.


**Summary Of The Paper:**

The paper is concerned with the scalability of training GNNs on large graph datasets where there may not be sufficient memory to store the graph structure. The basic idea is to decompose the graph into smaller components. The paper builds upon the earlier works such as (Chiang et al., 2019), which uses METIS to decompose a large graph into non-overlapping components and employs the alternating method to train the graph components and combine the results. In this paper, the authors basically also utilize the "residual" graph component (after the decomposition), the so-called "interface adjacency matrix", and construct a new (normalized) graph for simultaneous training. Experiment results show that the proposed method performs better than existing methods.

**Summary Of The Review:**

The paper builds upon the earlier works such as (Chiang et al., 2019), which uses METIS to decompose a large graph into non-overlapping components and employs the alternating method to train the graph components and combine the results. In this paper, the authors basically also utilize the "residual" graph component (after the decomposition), the so-called "interface adjacency matrix", and construct a new (normalized) graph for simultaneous training. While experiment results show that the proposed method performs better than existing methods, the proposed method is rather ad hoc without any mathematical justification. Perhaps more importantly, it requires a lot of more memory than existing methods based purely on  non-overlapping graph decomposition

---

### Decision · Program_Chairs · 2023-01-20

**Decision:**

Reject

**Justification For Why Not Higher Score:**

No baselines, little novelty, and high computational complexity.

**Justification For Why Not Lower Score:**

N/A

**Metareview: Summary, Strengths And Weaknesses:**

This work presents a method for improving the performance of Graph Convolutional Networks (GCN) on large graphs. The method involves decomposing the graph into non-overlapping subgraphs, training GCN on each subgraph, and combining the results using weighted averages. This allows GCN to be applied to each subgraph individually, reducing complexity. To mitigate the loss of information from ignoring the connections between subgraphs (known as "interface links"), the proposed method also trains GCN on the interface links and combines the results with those from the subgraphs. Reviewers thought that the work was generally easy to read, but could be improved by providing more motivation for the problem and explicitly stating some of the assumptions about the system. However, the paper does not include any analysis on the memory requirements or comparisons with other existing methods. Therefore, it is unclear if the proposed method is practical in terms of performance and memory usage compared to other methods.

- Reviewers were concerned with the absence of comparisons with existing methods, unclear novelty, and the high computational cost of the method.